

# Lagrangian process attribution of isotopic variations in near-surface water vapour in a 30-year regional climate simulation over Europe

Marina Dütsch[1], Stephan Pfahl[1], Miro Meyer[1], and Heini Wernli[1]

[1]Institute for Atmospheric and Climate Science, ETH Zurich, Zurich, Switzerland

*Correspondence to:* Marina Dütsch (marina.duetsch@env.ethz.ch)

**Abstract.** Stable water isotopes are naturally available tracers of moisture in the atmosphere. Due to isotopic fractionation, they record information about condensation and evaporation processes during the transport of air parcels, and therefore present a valuable means for studying the global water cycle. However, the meteorological processes driving isotopic variations are complex and not very well understood so far, in particular on short (hourly to daily) time scales. This study presents a Lagrangian method for attributing the isotopic composition of air parcels to meteorological processes, which provides new insight into the isotopic history of air parcels. It is based on the temporal evolution of the isotope ratios, the humidity, the temperature and the location of the air parcels. Here these values are extracted along seven-day backward trajectories started every six hours from near the surface in a 30-year regional climate simulation over Europe with the isotope-enabled version of the model of the Consortium for Small-Scale Modelling (COSMOiso). The COSMOiso simulation has a horizontal resolution of 0.25° and is driven at the lateral boundaries by a T106 global climate simulation with the isotope-enabled version of the European Centre Hamburg model (ECHAMwiso). Both simulations are validated against measurements from the Global Network of Isotopes in Precipitation (GNIP), which shows that nesting COSMOiso within ECHAMwiso improves the representation of $\delta^2$H and deuterium excess in monthly accumulated precipitation. The method considers all isotopic changes that occur inside the COSMOiso model domain, which, on average, correspond to more than half of the mean and variability of both $\delta^2$H and deuterium excess at the air parcels' arrival points. Along every trajectory, the variations of the isotope values are quantitatively decomposed into eight process categories (evaporation from the ocean, evapotranspiration from land, mixing with moister air, mixing with drier air, liquid cloud formation, mixed phase cloud formation, ice cloud formation, and no process). The results show that, for air parcels arriving over the ocean, evaporation from the ocean is the primary factor controlling $\delta^2$H and deuterium excess. Over land, evapotranspiration from land and mixing with moister air are similarly important. Liquid and mixed phase cloud formation contribute to the variability of $\delta^2$H and deuterium excess, especially over continental Europe. In summary, the presented method helps to better understand the linkage between the meteorological history of air parcels and their isotopic composition, and may support the interpretation of stable water isotope measurements in future.

## 1 Introduction

Stable water isotopes ($H_2^{16}O$, $HD^{16}O$ and $H_2^{18}O$) experience fractionation during phase transitions, meaning that they become enriched in one phase and depleted in the other. In this way they can record information about evaporation and condensation



processes during the transport of air parcels. Since the strength of fractionation depends on meteorological conditions (e.g., temperature, relative humidity and wind speed), stable water isotopes have become useful tracers of the global water cycle. For example, low $\delta^2$H or $\delta^{18}$O values in atmospheric water vapour (where the $\delta$ notation describes the concentrations of the heavy isotopes relative to Vienna Standard Mean Ocean Water – VSMOW) indicate low temperatures and strong rainout of air parcels

(e.g., Jacob and Sonntag, 1991; Yoshimura et al., 2011), and high deuterium excess values (defined as $d = \delta^2$H $- 8 \cdot \delta^{18}$O) indicate low relative humidities at the moisture sources (e.g., Gat et al., 2003; Aemisegger et al., 2014). Similar quantitative relations exist between atmospheric processes and isotope signals in precipitation (e.g., Dansgaard, 1964; Pfahl and Sodemann, 2014).

However, the attribution of isotope signals to individual meteorological processes is challenging, since all phase transitions

involving the vapour phase (except snow sublimation) cause fractionation and therefore change $\delta^2$H, $\delta^{18}$O and the deuterium excess. While $\delta^2$H and $\delta^{18}$O are to first order governed by equilibrium fractionation processes, which are caused by the higher binding energies of the heavy isotopes, the deuterium excess is more sensitive to nonequilibrium fractionation effects, which are caused by the slower diffusion velocities of the heavy isotopes. Hence, the isotope signal of an air parcel at a specific measurement site represents the total imprint of all equilibrium and nonequilibrium fractionation processes that occurred during

its transport, typically including evaporation from the ocean, evapotranspiration from land, cloud formation, and below-cloud rain evaporation and equilibration. Additionally, mixing with surrounding air can further influence the isotopic composition of air parcels. These processes can occur almost simultaneously, and most air parcels experience a combination of them on short time scales of a few hours to days. Thus, to fully explore the potential of stable water isotopes as tracers of the water cycle a good understanding of the imprint of these processes on the isotopic composition of air parcels is necessary.

Due to this complexity, numerical models are an essential tool for studying these processes and their influence on isotopes. Several Eulerian and Lagrangian isotope models have been developed so far. Eulerian isotope models simulate the atmosphere (and oceans) by solving the equations expressing conservation of momentum, energy, and mass on a fixed grid, and represent, in parts with simplified parameterisations, all meteorological processes that modify the isotopic composition of water vapour and precipitation. They have been used for sensitivity studies to clarify the role of specific processes on isotopic variability.

For example, Risi et al. (2013) used the isotope-enabled version of the Laboratoire de Météorologie Dynamique Zoom general circulation model (LMDZiso; Risi et al., 2010) to quantify the effects of moisture source conditions, rainout, mixing, rain reevaporation and supersaturation in ice clouds on the deuterium excess and the $^{17}$O excess in precipitation at different latitudes. Moore et al. (2014) used the isotope version of the System for Atmospheric Modeling (IsoSAM; Blossey et al., 2010) to determine the relative importance of moisture convergence and rain evaporation and equilibration for the amount effect (i.e.,

decreasing $\delta^2$H and $\delta^{18}$O with increasing precipitation amount) in an idealised simulation, and Christner et al. (2017b) used the isotope version of the Consortium for Small-Scale Modelling (COSMOiso; Pfahl et al., 2012) for attributing $\delta^2$H in European water vapour and precipitation to evapotranspiration, rainout, and rain evaporation and equilibration. The disadvantage of Eulerian isotope models is that, due to their complexity and consequently the many inherent feedbacks, it can be difficult to isolate the impact of individual processes on isotopic variability.

In contrast, Lagrangian isotope models follow air parcels and simulate their isotopic composition during transport. They pro-



vide a direct link to the most important processes and moisture sources, while still having a relatively simple numerical structure. They have been applied for understanding the isotopic history of air parcels arriving at various measurement sites. For example, Helsen et al. (2007) used a Rayleigh-type isotope model in combination with backward trajectory calculations to simulate and interpret the isotopic composition of snow in Antarctica. Sodemann et al. (2008a) applied a Lagrangian moisture

source diagnostic to simulate the effects of the North Atlantic Oscillation on $\delta^2$H and $\delta^{18}$O in snow on Greenland. Sinclair et al. (2011) used three different Rayleigh-type models to test the effect of topography on $\delta^{18}$O in snow in western Canada, and Christner et al. (2017a) used a Lagrangian isotope model based on realistic trajectories to quantify the influence of snow sublimation and meltwater evaporation on $\delta^2$H in water vapour over Europe. The disadvantage of Lagrangian isotope models is that, since they require a strong simplification of meteorological processes, e.g., they neglect mixing with surrounding air

(Noone and Sturm, 2010), they cannot capture the full complexity of the global water cycle.

In this study, we combine the Eulerian and Lagrangian approaches by tracing isotopes along Lagrangian backward trajectories in a 30-year Eulerian model simulation over Europe. Thus, we avoid some of the simplifications used in Lagrangian isotope models, while still focusing on the history of air parcels and on the imprint of meteorological processes on their isotopic composition. With the help of this new approach we will address the following question: Which processes in the atmospheric

water cycle determine the mean and variability of the isotope signal in near-surface water vapour at different locations across Europe?

The long-term simulation of 30 years allows investigating variability from the daily to the interannual time scale. We focus on water vapour near the surface, where isotopes are most often measured, and distinguish between the following seven process categories: evaporation from the ocean, evapotranspiration from land, mixing with moister air, mixing with drier air, liquid

cloud formation, mixed phase cloud formation, and ice cloud formation. For the simulation the limited area isotope model COSMOiso (Pfahl et al., 2012) is used, which is based on the nonhydrostatic weather forecast and climate model COSMO (Steppeler et al., 2003). Concentrating on a limited area allows a high output frequency and a high spatial resolution (here 0.25°), which is favourable for trajectory calculations. A new method is introduced to attribute all isotopic changes along the trajectories to one of the meteorological processes mentioned above, or to a "no process category", as will be explained in

Section 3.2. In this way the contribution of each process to the air parcels' final isotopic composition can be determined. This evaluation with a large set of trajectories driven by a high-resolution regional climate simulation is a promising new approach to address the above question in a quantitative way. The results will be presented in Sections 3.5 and 3.6 and discussed in Section 4. Conclusions are given in Section 5.

## 2   30-year COSMOiso simulation over Europe

### 2.1   Model

COSMO (Steppeler et al., 2003) is a numerical weather forecast and climate model that is operationally used at several European weather services. It is based on the hydro-thermodynamical equations describing compressible nonhydrostatic flow and can be used for simulations with horizontal resolutions of less than 1 km. The isotope implementation (COSMOiso; Pfahl et al.,



2012) is similar to other Eulerian isotope models (e.g., Joussaume et al., 1984; Sturm et al., 2005; Blossey et al., 2010; Werner et al., 2011): it includes two parallel water cycles for each of the heavy isotopes ($HD^{16}O$ and $H_2^{18}O$), which are used purely diagnostically and do not affect other model components. The heavy isotopes experience the same processes as the light isotope ($H_2^{16}O$), except during phase transitions, when isotopic fractionation occurs. A one-moment microphysics scheme is used and

convection is parameterised following Tiedtke (1989). For a detailed description of the physics and isotope parameterisations see Doms et al. (2011) and Pfahl et al. (2012), respectively.

## 2.2 Simulation setup

The simulation is run for 30 years (1982 – 2011) in a model domain covering most of Europe, the Mediterranean, and part of the North Atlantic and North Africa (Figure 1). The grid spacing is $0.25°$ in the horizontal and between 16 m and 2808 m in

the vertical, with 40 terrain following model levels. Initial and boundary conditions are provided by a nudged historical isotope simulation (Butzin et al., 2014) performed with the isotope-enabled European Centre Hamburg Model (ECHAMwiso; Werner et al., 2011) at T106 horizontal resolution and on 31 vertical levels. COSMOiso runs freely inside the model domain. Isotopic fractionation during evaporation from the ocean is parameterised with the Craig-Gordon model (Craig and Gordon, 1965) using a wind speed independent formulation of the nonequilibrium fractionation factor (Pfahl and Wernli, 2009). No fractionation

is assumed to occur during evapotranspiration from land surfaces, and the isotope content of the soil is prescribed by external data from ECHAMwiso. This setup has been found to be the best out of six setups when compared with monthly isotope measurements in precipitation from the Global Network of Isotopes in Precipitation (GNIP; IAEA/WMO, 2016) (Dütsch, 2016).

## 2.3 Evaluation with GNIP measurements

Figure 2 shows $\delta^2H$ and deuterium excess in precipitation in ECHAMwiso and COSMOiso together with GNIP measurements. The values correspond to the unweighted mean of the weighted monthly means, i.e., the monthly means are weighted by the precipitation amount (according to the GNIP measurements), but the 30-year mean of the models and the measurements is unweighted. Only stations that measured during at least a third of the time are considered. Both models nicely reproduce the $\delta^2H$ gradients from the continent to the ocean (continental effect) and from north to south (latitude effect), which are also observed

at the GNIP stations (Figure 2 a). The finer resolution of the COSMOiso simulation, and consequently its better representation of topography, reveals small-scale structures of $\delta^2H$ that are not visible in ECHAMwiso. For example, the Atlas, Pyrenees, Carpathians or Balkan mountains clearly receive more depleted precipitation than their surrounding flatlands. Furthermore, COSMOiso tends to produce more depleted precipitation than ECHAMwiso, especially over continental Europe. The deuterium excess varies on small horizontal scales and differs more between the models and the GNIP measurements (Figure 2 b).

This indicates that nonequilibrium fractionation, which to first order governs the deuterium excess, is more difficult to simulate than equilibrium fractionation, which to first order governs $\delta^2H$ and $\delta^{18}O$. ECHAMwiso produces the highest deuterium excess over North Africa, while COSMOiso has the highest values over the Mediterranean and relatively low values everywhere else. In Figure 3 the modelled 30-year averaged seasonal values of $\delta^2H$ and deuterium excess in precipitation are plotted against the





GNIP measurements (again the unweighted mean of the weighted monthly means). The modelled values are interpolated linearly to the GNIP stations. At all stations only the months are considered, for which a GNIP measurement and values from both models are available. Furthermore, only stations are shown with measurements in at least a third of the months in the respective season and 30 years. As already indicated in Figure 2, ECHAMwiso tends to overestimate and COSMOiso tends to underestimate $\delta^2$H in precipitation (Figure 3 a). Nonetheless the mean bias error (MBE) as well as the mean absolute error (MAE) of both models are small and the correlations ($R$) are high (0.92 for ECHAMwiso, 0.94 for COSMOiso). COSMOiso slightly outperforms ECHAMwiso in terms of all three measures for $\delta^2$H. The correlations of modelled and measured deuterium excess are smaller (0.65 for ECHAMwiso, 0.68 for COSMOiso; Figure 3 b). ECHAMwiso overestimates low and underestimates high deuterium excess values, and COSMOiso does the opposite. This leads to a relatively small MBE in both models, but a larger MAE, while ECHAMwiso outperforms COSMOiso in terms of MAE.

Figure 4 depicts the statistical distributions of the individual monthly $\delta^2$H and deuterium excess values of the two models and the GNIP measurements, again only for the months for which a GNIP measurement and values from both models are available. The probability density functions were calculated with a kernel density estimate using Gaussian kernels and Scott's rule (Scott, 1992) for the bandwidth selection. The distribution of $\delta^2$H is left-skewed (Figure 4 a), whereas the deuterium excess is almost normally distributed (Figure 4 b). COSMOiso nicely matches the measured distributions of both $\delta^2$H and deuterium excess with only a small shift towards lower values. ECHAMwiso produces too narrow distributions and overestimates their medians. Overall, both $\delta^2$H and deuterium excess from the GNIP measurements are better reproduced by COSMOiso than by ECHAMwiso. This underlines the added value of high resolution numerical model simulations of stable water isotopes, and motivates the following more detailed analysis of the isotopic signals in near-surface atmospheric water vapour induced by the different meteorological processes.

## 3 Isotopic history of air parcels

### 3.1 Trajectories

To represent the history of air parcels, stable water isotopes are traced along backward trajectories. The trajectories are computed from one hourly output fields of the COSMOiso simulation using the Lagrangian Analysis Tool (LAGRANTO; Sprenger and Wernli, 2015). They start every six hours and every 0.5° (Figure 1) from the first, third and fifth model level and go seven days backward in time or until they leave the model domain. Along the trajectories, one out of eight process categories (evaporation from the ocean, evapotranspiration from land, mixing with moister air, mixing with drier air, liquid cloud formation, mixed phase cloud formation, ice cloud formation, no process) is assigned to every time step, based on the change in specific humidity, the location, temperature and relative humidity of the trajectory. The allocation of the process categories builds upon the moisture source analysis introduced by Sodemann et al. (2008b) and is described in the following.





## 3.2 Process allocation

If specific humidity increases during a certain time step ($\Delta q > \Delta q_{min}$), moisture is assumed to have evaporated from the surface or to be mixed into the air parcel (i.e., the parcel mixes with moister air). The distinction between evaporation and mixing is made based on the trajectory's height $z$ compared to the boundary layer height BLH and the sign of the hourly accumulated evaporation flux $E$ from the surface. If the trajectory is located within an extended boundary layer ($z \leq 1.5 \cdot$ BLH, where the factor 1.5 takes into account the uncertainty of the boundary layer height parameterisation in COSMO), and the evaporation flux is from the surface to the atmosphere ($E > 0$), the moisture increase is assigned to evaporation, if not it is assigned to mixing. Evaporation is said to originate from the ocean if the interpolated land sea mask LSM at the trajectory's location is $\leq 0.75$, and from land otherwise. Note that there is no separate category for rain evaporation. This process is included either in evaporation from the surface or mixing depending on the trajectory's location.

If specific humidity decreases ($\Delta q < -\Delta q_{min}$), moisture is assumed to have condensed or to be mixed out of the air parcel (i.e., the parcel mixes with drier air). The distinction between condensation and mixing is made based on the relative humidity $h$ with respect to liquid water in the air parcel. If $h \geq 80\%$, the probability for subgrid-scale condensation is high, and the moisture decrease is assigned to condensation, if not it is assigned to mixing. The condensate is liquid if temperature $T > 0°$C, solid (ice) if $T < -23°$C, and mixed phase if $-23°$C $\leq T \leq 0°$C.

To avoid noise, a minimum change of specific humidity $\Delta q_{min} = 0.01\,\mathrm{g\,kg}^{-1}$ is required. If $-\Delta q_{min} \leq \Delta q \leq \Delta q_{min}$, no process is assigned to the time step. The results of this process allocation can be seen exemplarily in Figure 5. For instance, at $t = 60\,$h, specific humidity decreases while the trajectory is ascending above the boundary layer. Temperature is below $0°$C and the relative humidity is above $80\%$, therefore the attributed process is condensation in mixed phase clouds. During an extended period around $t = 120\,$h, specific humidity increases while the trajectory is in the oceanic boundary layer and ocean evaporation is the attributed process. A third example is the moisture increase at $t = 160\,$h, when the trajectory is in the boundary layer over land. In this case the relevant process is land evapotranspiration. An idealisation here is that only one process, the one that is regarded as most important, is attributed to a time step along the trajectories. In reality, different processes might often act simultaneously.

## 3.3 Weighting with moisture

An air parcel typically experiences several moisture uptakes and losses. Processes occurring earlier during the transport of the air parcel therefore contribute less to its final isotopic composition, since part of the signal is lost during moisture losses and overwritten by later moisture uptakes. The relative contribution of a process at time $n$ is proportional to the amount of moisture $q_{fin}^n$ at time $n$ that is still contained in the air parcel at its arrival point (at time $N$). To determine $q_{fin}^n$ the corresponding fraction of moisture $f_{fin}^n = q_{fin}^n / q^n$ is calculated. With $f_{fin}^N = 1$ by definition, $f_{fin}^n$ along the trajectories can be derived backward in





time:

$$f_{fin}^n = f_{fin}^{n+1} \cdot \min(q^{n+1}/q^n, 1) \tag{1}$$

$$q_{fin}^n = f_{fin}^n \cdot q^n \tag{2}$$

where $\min(q^{n+1}/q^n, 1)$ denotes the minimum of 1 and the ratio of specific humidities at times $n+1$ and $n$. This weighting is equivalent to the weighting applied in the moisture source diagnostic by Sodemann et al. (2008b). Figure 6 shows the time series of $q$, $q_{fin}$, and $f_{fin}$ along a hypothetical trajectory. If $q$ decreases from time $n$ to time $n+1$ (forward), $q_{fin}$ stays constant, while $f_{fin}$ increases. This means that the contributions of the times $n$ and $n+1$ to the air parcel's final composition are equal, since no new moisture enters the parcel from time $n$ to $n+1$. If $q$ increases, $q_{fin}$ increases proportionally, while $f_{fin}$ stays constant. This means that the contribution of time $n+1$ is larger than the contribution of time $n$, since the air parcel takes up new moisture, which partly overwrites the signal from time $n$. If $q$ stays constant, both $q_{fin}$ and $f_{fin}$ stay constant as well.

### 3.4 Isotopes

For quantifying the impact of the processes on the air parcels' final isotope ratio $\delta^N$ (where $\delta = \delta^2\text{H}$ or $d$), we consider, at time $n$, only the moisture that is still contained in the air parcel at its arrival point ($q_{fin}^n$), and reformulate $\delta^N$ to express the sum of the initial isotope ratio $\delta^0$ and the changes during transport, all weighted by $q_{fin}^n$:

$$q_{fin}^N \cdot \delta^N = q_{fin}^0 \cdot \delta^0 + \left(q_{fin}^1 \cdot \delta^1 - q_{fin}^0 \cdot \delta^0\right) + ... + \left(q_{fin}^N \cdot \delta^N - q_{fin}^{N-1} \cdot \delta^{N-1}\right) \tag{3}$$

$$= q_{fin}^0 \cdot \delta^0 + \sum_{n=0}^{N-1}\left(q_{fin}^{n+1} \cdot \delta^{n+1} - q_{fin}^n \cdot \delta^n\right) \tag{4}$$

$$= q_{fin}^0 \cdot \delta^0 + \sum_{n=0}^{N-1} \Delta\left(q_{fin}^n \cdot \delta^n\right) \tag{5}$$

Hence, the air parcels' final isotope ratio ($\delta_{fin} \equiv \delta^N$) is given by the sum of the weighted initial isotope ratio ($\delta_{ini}$) and the weighted changes along the trajectories ($\Delta\delta$):

$$\delta_{fin} = \underbrace{\frac{q_{fin}^0}{q_{fin}^N} \cdot \delta^0}_{\delta_{ini}} + \underbrace{\sum_{n=0}^{N-1} \frac{\Delta\left(q_{fin}^n \cdot \delta^n\right)}{q_{fin}^N}}_{\Delta\delta} \tag{6}$$

where $\delta_{ini}$ corresponds to the value of the trajectory either seven days prior to its arrival or at the time when it enters the model domain. Furthermore, the contribution of a process $k$ to the final isotopic composition is:

$$\Delta\delta_k = \sum_{n_k=0}^{N_k-1} \frac{\Delta\left(q_{fin}^{n_k} \cdot \delta^{n_k}\right)}{q_{fin}^N} \quad \text{with} \quad \sum_{k=1}^{8} \Delta\delta_k = \Delta\delta \quad \text{and} \quad \sum_{k=1}^{8} N_k = N \tag{7}$$

where the subscript $k$ denotes time steps assigned to process $k$.

For the processes that increase the moisture content of the air parcels ($q_{fin}^{n_k+1} > q_{fin}^{n_k}$), $\Delta\delta_k$ corresponds to the isotopic compo-




sition of the moisture added by the processes, weighted by $\Delta q_{fin}^{n_k}/q_{fin}^N$:

$$\Delta\delta_k = \sum_{n_k=0}^{N_k-1} \left( \underbrace{\frac{\Delta q_{fin}^{n_k}}{q_{fin}^N}}_{\text{weight}} \cdot \underbrace{\frac{\Delta\left(q_{fin}^{n_k}\cdot\delta^{n_k}\right)}{\Delta q_{fin}^{n_k}}}_{\delta \text{ in moisture uptake}} \right) \tag{8}$$

For the processes that decrease the moisture content of the air parcels ($q_{fin}^{n_k+1} = q_{fin}^{n_k}$), $\Delta\delta_k$ corresponds to the change in isotopic composition experienced by the remaining moisture due to the processes, weighted by $q_{fin}^{n_k}/q_{fin}^N$.

$$\Delta\delta_k = \sum_{n_k=0}^{N_k-1} \left( \frac{q_{fin}^{n_k}\cdot\delta^{n_k+1} - q_{fin}^{n_k}\cdot\delta^{n_k}}{q_{fin}^N} \right) = \sum_{n_k=0}^{N_k-1} \left( \underbrace{\frac{q_{fin}^{n_k}}{q_{fin}^N}}_{\text{weight}} \cdot \underbrace{\Delta\delta^{n_k}}_{\text{change of }\delta} \right) \tag{9}$$

Comparison of the different $\Delta\delta^2\mathrm{H}_k$ and $\Delta d_k$ will then show which processes contributed how much to the final isotopic composition of the air parcels.

The summed up weights of each process will hereafter be referred to as the amount of moisture "explained" by the process (in %):

$$q_{exp}^k = \sum_{n_k=0}^{N_k-1} \frac{\Delta q_{fin}^{n_k}}{q_{fin}^N} \quad \text{for moisture increasing processes} \tag{10}$$

$$q_{exp}^k = \frac{1}{N} \cdot \sum_{n_k=0}^{N_k-1} \frac{q_{fin}^{n_k}}{q_{fin}^N} \quad \text{for moisture decreasing processes} \tag{11}$$

The sum of all $q_{exp}^k$ of the moisture increasing processes will be referred to as the total fraction of explained moisture ($q_{exp}$) and corresponds to the fraction of final moisture taken up by the trajectories inside the model domain ($q_{exp} = 1 - q_{fin}^0/q_{fin}^N$). The influence of the processes on isotopic variability is addressed by considering days, months and years when $\delta^2\mathrm{H}$ or deuterium excess are unusually high or low, i.e., when the anomalies with respect to the climatological mean are in the highest or lowest 25 % (33 % for years). This means that, for each grid point, the 2739 days, 90 months, and 10 years with the highest and lowest $\delta^2\mathrm{H}$ and deuterium excess anomalies are selected. For the daily anomalies, the climatological mean is calculated as the 31-day running mean of the 30-year daily climatology, for the monthly anomalies it corresponds to the 30-year monthly climatology, and for the yearly anomalies to the 30-year yearly climatology. The difference of $\Delta\delta^2\mathrm{H}_k$ and $\Delta d_k$ between high and low anomaly days, months and years represents the contribution of process $k$ to the anomalies (and thus, to isotopic variability on the given time scale).

## 3.5  Mean $\delta^2\mathrm{H}$ and deuterium excess

Figure 7 shows the 30-year mean $\delta^2\mathrm{H}$ and deuterium excess in water vapour averaged over the first, third and fifth lowest model level ($\delta^2\mathrm{H}_{fin}$ and $d_{fin}$, Figure 7 a,b) together with the separate contributions of the initial values ($\delta^2\mathrm{H}_{ini}$ and $d_{ini}$,



Figure 7 c,d) and the total changes along the trajectories induced by the processes specified in Section 3.2 ($\Delta\delta^2$H and $\Delta d$, Figure 7 e,f). By construction, the weighted initial values and changes along the trajectories fully explain $\delta^2$H and $d$ at the trajectories' arrival points ($\delta_{ini} + \Delta\delta = \delta_{fin}$; see Equation 6). Note that $\delta_{ini}$ and $\Delta\delta$ are not drawn at the grid points where they occur, but at the grid points where the corresponding trajectory arrives. Also, $\delta_{fin}$ corresponds to a straightforward Eu-

lerian average of the COSMOiso output over 30 years and model levels 1,3 and 5. The percentage of moisture explained by the initial state and the changes along the trajectories is shown in contours and corresponds to $1 - q_{exp} = q_{fin}^0 / q_{fin}^N$ for the initial state (Figure 7 c,d) and to $q_{exp}$ for the changes along the trajectories (Figure 7 e,f). Due to the multiplication of $\delta^2$H with $q_{fin}^n$ at each time step (see Equation 3), the patterns of $\delta_{ini}$ and $\Delta\delta$ depend on both $q_{exp}$ and $\delta$. For $\delta^2$H, which is (almost) always negative, multiplication with a higher positive number (larger $q_{exp}$) results in a lower (more negative) number. Thus

a lower $\Delta\delta^2$H can result from a lower $\delta^2$H in moisture uptakes, a stronger decrease/weaker increase of $\delta^2$H during moisture losses, or a larger $q_{exp}$ (or a combination of the three). For deuterium excess, which is typically positive, multiplication with a higher positive number results in a higher (more positive) number, and a higher $\Delta d$ can result from a higher deuterium excess in moisture uptakes, a stronger increase/weaker decrease of deuterium excess during moisture losses or a larger $q_{exp}$ (or again a combination of the three).

Similarly as $\delta^2$H in precipitation (cf. Figure 2 a), $\delta^2$H$_{fin}$ in water vapour shows a positive gradient from north to south (latitude effect) and from the continent towards the ocean (continental effect) (Figure 7 a). The latitude effect is already visible in $\delta^2$H$_{ini}$ (Figure 7 c). However this is mainly due to the lower $q_{exp}$ in the north than in the south, meaning that more initial moisture is contained in the air parcels in the north than in the south. $\delta^2$H$^0$ alone has no north-south gradient (not shown). The processes ($\Delta\delta^2$H; Figure 7 e) add the land-sea contrast (continental effect) to $\delta^2$H$_{fin}$ and the depletion in mountainous regions

(altitude effect). $d_{ini}$ (Figure 7 d) shows a similar pattern as $\delta^2$H$_{ini}$, but of the opposite sign. This again corresponds quite well to the pattern of $1 - q_{exp}$. The high values of $d_{fin}$ (Figure 7 b) in the south of the domain, especially over the Mediterranean, originate from the changes along the trajectories (Figure 7 f). The mean contribution of $\Delta\delta^2$H to $\delta^2$H$_{fin}$ and $\Delta d$ to $d_{fin}$ are 60 % and 73 %, respectively, meaning that more than half of the mean of both isotope parameters is determined during the previous seven days and within the COSMOiso domain. These parts can be further separated into the relative contributions of

the different processes. They are shown in Figure 8 for $\delta^2$H and in Figure 9 for deuterium excess. For $\delta^2$H all contributions are negative, since $\delta^2$H in the moisture added to the air parcels, e.g., by surface evaporation, is (almost) always negative, and moisture decreasing processes, such as rainout, typically also decrease $\delta^2$H. For deuterium excess the contributions are mostly positive but can also be negative, since the deuterium excess in the moisture uptakes is typically positive, but some processes, such as the formation of liquid clouds, lead to a decrease.

For $\delta^2$H, evaporation from the ocean contributes most with 37 % on average. Mixing with moister air, evapotranspiration from land, mixing with drier air, and the formation of liquid clouds contribute with 20 %, 17 %, 12 % and 11 % respectively. The formation of mixed phase and ice clouds, and the time steps for which no process was assigned are only of minor importance. For the trajectories arriving over the ocean, evaporation from the ocean is the dominant process. Over land, evapotranspiration from land, mixing with moister and with drier air, and the formation of liquid clouds are more important than evaporation from

the ocean. The patterns correspond quite well to the amount of moisture explained by the processes ($q_{exp}^k$). It is generally high





where $\delta^2$H is low, meaning that the importance of the processes for determining $\delta^2$H is in line with the amount of moisture they contribute.

For deuterium excess, evaporation from the ocean is clearly the most important process with $65\,\%$ contribution on average. The other two moisture increasing processes, evapotranspiration from land and mixing with moister air, account for $20\,\%$ and $16\,\%$,

respectively. Cloud formation processes and mixing with drier air have almost no influence. The formation of liquid clouds shows a small negative contribution $(-3\,\%)$, meaning that it slightly decreases the deuterium excess. Again, evaporation from the ocean is most important for the trajectories arriving over the ocean, while for the trajectories arriving over land evapotranspiration from land and mixing with moister air are dominant. The contribution of the moisture increasing processes to the final deuterium excess also corresponds well to their respective $q_{exp}^k$, whereas no relation to $q_{exp}^k$ can be found for the moisture

decreasing processes.

### 3.6    Variability of $\delta^2$H and deuterium excess

Figure 10 shows the difference in the anomalies of $\delta^2$H with respect to the climatological mean between the $25\,\%$ of days and months and $33\,\%$ of years when they were highest and the $25\,\%$ of days and months and $33\,\%$ of years when they were

lowest. Stippling indicates where the probability of obtaining the difference between the two samples (high and low) by chance is $p < 1\,\%$. $p$ was calculated from a two-sided t-test for the null hypothesis that the samples have identical mean values. The difference in $\delta^2$H is larger over land than over ocean with values above $38\,\%o$ for the daily, above $19\,\%o$ for the monthly, and above $4.75\,\%o$ for the yearly anomalies (Figure 10 a). This means that the variability of $\delta^2$H is larger over land than over ocean. Apart from the scale divided by 2 and 8 for the monthly and yearly compared to the daily anomalies, the three spatial patterns are very similar. Also here, the pattern of $\Delta\delta^2$H depends on both $q_{exp}$ and $\delta^2$H. Thus, a positive anomaly of $\Delta\delta^2$H can imply

a higher $\delta^2$H and/or a lower $q_{exp}$.

For these anomalies the contribution of the initial values ($\delta^2\mathrm{H}_{ini}$; Figure 10 b) is smaller and the contribution of the changes along the trajectories ($\Delta\delta^2$H; Figure 10 c) is larger than for the climatological mean $\delta^2\mathrm{H}_{fin}$ (cf. Figure 7). The changes along the trajectories on average account for $66\,\%$, $75\,\%$, and $62\,\%$ of the daily, monthly and yearly anomalies, respectively. This

means that they are mainly responsible for the observed variability of $\delta^2$H. The $\delta^2$H anomalies resulting from the changes along the trajectories can again be separated into the different processes (Figure 11). The results for the daily, monthly and yearly anomalies are very similar, and we therefore only show results for the daily anomalies here (see Supplementary Material Figures S1 and S2 for the monthly and yearly anomalies, respectively). The contributions of the different processes show a large spatial variability. Over the ocean the main reason for the higher $\delta^2$H are the higher $\delta^2$H values during evaporation from

the ocean, since the difference in $q_{exp}^k$ is small (Figure 11 d). Evaporation from the ocean leads to a negative anomaly at the coasts of the Mediterranean, Black Sea, and partly the North Atlantic. Here this is due to the higher fraction of moisture coming from the ocean on days with high $\delta^2$H (and not due to lower $\delta^2$H, as can be seen from the positive difference in $q_{exp}^k$ from ocean evaporation in these regions). For the same areas the fraction of moisture coming from evapotranspiration from land is lower, resulting in a positive anomaly for this process (Figure 11 b), which compensates the negative anomaly from evaporation from





the ocean. Over the Atlas mountains the negative anomaly is compensated by mixing with moister air (Figure 11 f). Hence these trajectories experience more evaporation from the ocean and less evapotranspiration from land and mixing on days with high $\delta^2$H. Over other parts of Europe the high $\delta^2$H anomalies are a mixture of contributions from mixed phase and liquid cloud formation (less mixed phase and more liquid clouds associated with weaker fractionation and thus higher $\delta^2$H), evaporation

from the ocean, evapotranspiration from land, and mixing with moister air. Especially noteworthy is the large contribution of mixed phase clouds to the $\delta^2$H anomalies (on average 15 %; Figure 11 c) in comparison to their small influence on the climatological mean $\delta^2$H (on average 3 %; see Figure 8 c). Mixing with drier air is mostly relevant over North Africa. Ice clouds have no influence, and the time steps for which no process was assigned contribute slightly positively to the anomalies over Europe.

Figure 12 shows the difference between the high and low anomaly composites for deuterium excess. Here the difference is larger over ocean than over land, especially on the daily time scale (Figure 12 a), hence, the variability is larger over the ocean. The initial values contribute slightly negatively to the daily, monthly and yearly deuterium excess anomalies (Figure 12 b). As a consequence, the changes along the trajectories contribute with more than 100 % to the anomalies (Figure 12 c), and are therefore able to fully explain the variability of deuterium excess. Note that the negative contribution from the initial values

indicates that the deuterium excess from evaporation inside the domain is typically larger than outside of the domain. This might be related to the different parameterisations of nonequilibrium fractionation during evaporation from the ocean in the driving model ECHAMwiso compared to COSMOiso.

Separation of these changes into the different processes shows less spatial variability than for the $\delta^2$H anomalies, and a clear dominance of one process, which is evaporation from the ocean (Figure 13 for the daily anomalies, Figures S3 and S4 for the

monthly and yearly anomalies, respectively). It contributes with 72 % on average to the daily anomalies, and with 82 % over the ocean. This is primarily due to the larger moisture input from the ocean (positive $q_{exp}^k$ difference in Figure 13 d) due to lower relative humidity and stronger evaporation (which is likely related to the lower amount of liquid clouds over the ocean, i.e., the negative $q_{exp}^k$ difference in Figure 13 a). Over land the formation of liquid clouds, evapotranspiration from land, and mixing with moister air are similarly important as evaporation from the ocean. Mixed phase clouds, ice clouds, mixing with

drier air, and the time steps, for which no process is assigned, are of minor importance for deuterium excess variability.

## 4 Discussion

This study presents a new approach for attributing the isotopic composition of water vapour to meteorological processes, which can help interpretating isotope measurements at different locations across Europe. The two isotope parameters $\delta^2$H and deuterium excess were traced along backward trajectories in a 30-year climatological COSMOiso simulation. COSMOiso

reproduced the mean and variability of $\delta^2$H in GNIP measurements remarkably well, and led to an improvement compared to the global model ECHAMwiso. This is in accordance with previous studies that demonstrated the added value of regional high-resolution simulations of stable water isotopes (e.g., Sturm et al., 2007; Pfahl et al., 2012). Here we limited the evaluation to GNIP measurements in precipitation, and therefore cannot completely rule out compensating errors of different fractionation





processes. However, Christner et al. (2017b) validated a similar climatological simulation with COSMOiso over Europe against measurements in water vapour as well as GNIP measurements and found that the model reproduces $\delta^2$H in water vapour even more accurately than $\delta^2$H from the GNIP measurements. Thus we are confident that COSMOiso represents the influence of the different meteorological processes on $\delta^2$H in a realistic way. The deuterium excess, being more sensitive to nonequilib-

rium fractionation processes, is more difficult to simulate than $\delta^2$H and the correlations between the modelled and measured values were lower. This means that the imprint of meteorological processes on deuterium excess could potentially be biased and therefore the results for deuterium excess have to be taken more cautiously. Nevertheless, the nicely matched statistical distribution of the modelled and measured values shows that COSMOiso is able to cover a representative range of deuterium excess values.

The decomposition of the $\delta^2$H and deuterium excess signals into the different processes showed that the moisture increasing processes (ocean evaporation, land evapotranspiration, mixing with moister air) mainly determine $\delta^2$H and deuterium excess in water vapour at the lowest model levels, while the moisture decreasing processes (formation of liquid clouds, mixed phase clouds, ice clouds and mixing with drier air) had smaller contributions, although some of them (e.g., the formation of mixed phase clouds) were important for the variability. Ocean evaporation and land evapotranspiration were dominant especially for

the deuterium excess, which is in line with its common application as a proxy for moisture source conditions (e.g., Jouzel et al., 1982; Steffensen et al., 2008; Aemisegger et al., 2014). In the current setting with the starting points of the backward trajectories at low levels, all air parcels descend towards the end (except those having their starting point on a mountain). This means that they tend to take up moisture towards the end, which overwrites previous signals from other processes. This may explain why the moisture increasing processes are more important than the moisture decreasing processes. In future research

we will investigate how the results change for trajectories started from higher levels.

A limitation of the study is the binary distinction between processes based on thresholds, which assumes that only one process occurs per time step. With a time step of one hour, this is not very realistic. For example, it could happen that an air parcel loses moisture by precipitation but at the same time takes up new moisture from evaporation. This would mean that either the contribution of cloud formation or of evaporation is underestimated, depending on the net change in specific humidity. At the

same time it would lead to an overestimation of processes occurring earlier during the transport of the air parcel, since they are not discounted during the hidden moisture loss. This has been shown in a similar way by Beusch (2017), who performed a moisture source analysis with trajectories based on different temporal resolutions and found a tendency towards more local moisture sources with increasing temporal resolution due to larger fluctuations in specific humidity and stronger discounting of earlier moisture uptakes. However, also the amount of noise increases with increasing temporal resolution, leading to un-

realistically many moisture uptakes and losses on short time scales. Thus, it is not sure whether a higher temporal resolution would improve the estimate of the contributions of the different meteorological processes. The sensitivity of the results to the selection of thresholds (specifically LSM and $h$) has been tested for one year and two locations by Meyer (2016). As can be expected, changing the LSM threshold from LSM $\leq 0.75$ to LSM $\leq 0.5$ leads to a larger contribution of land evapotranspiration and a smaller contribution of ocean evaporation, while changing the $h$ threshold from $h \geq 80\,\%$ to $h \geq 70\,\%$ leads to a larger

contribution of cloud formation and a smaller contribution of mixing with drier air. However, the differences are small, and the




patterns are qualitatively very similar. We abstained from performing sensitivity tests in the 30-year simulation, not least due to limited computational resources.

Furthermore, one aspect that is currently not accounted for is rain evaporation and equilibration. This process is especially important for the isotopic composition of precipitation, however it can also influence water vapour (e.g., Aemisegger et al.,

2015). In future research the distinction between rain evaporation and equilibration and mixing or evaporation from the surface could be made by additionally tracing liquid cloud water and rain along the trajectories, and categorising a time step as rain evaporation and equilibration if rain is present but no liquid cloud water. However, for this purpose a higher temporal resolution of the trajectories would be required.

## 5   Conclusions

We have presented a Lagrangian process attribution of isotopic variations in water vapour, which follows air parcel trajectories and assigns all isotopic changes during transport to a meteorological process. In this way, we quantified the imprint of the processes on the isotopic composition of near surface water vapour in a 30-year climatological simulation using the regional isotope-enabled model COSMOiso with lateral boundary conditions from ECHAMwiso. The main findings of the study can be summarised as follows:

1. Nesting COSMOiso within ECHAMwiso improves the representation of $\delta^2$H and deuterium excess when validated against GNIP measurements, underlining the added value of simulating stable water isotopes with high spatial resolutions.

      2. More than half of the mean and variability of $\delta^2$H and deuterium excess in near surface water vapour can be explained with the help of seven day backward trajectories, meaning that they are determined by processes occurring during the
20       previous seven days and within the simulation domain.

      3. For trajectories arriving over the ocean, evaporation from the ocean is the primary factor controlling $\delta^2$H and deuterium excess. Over land, evapotranspiration from land and mixing with moister air are similarly important.

      4. Liquid and mixed phase cloud formation contribute to the variability of $\delta^2$H and deuterium excess, especially over continental Europe.

The Lagrangian process attribution is a new method for better understanding the meteorological history of air parcels arriving at a measurement site. It provides new insight into the characteristics of the different processes that influence the isotopic composition of air parcels during transport, and may support the interpretation of isotope measurements in water vapour (and precipitation) in future studies.

## 6   Data availability

COSMOiso output data are available from the authors upon request (marina.duetsch@env.ethz.ch).



*Acknowledgements.* This work was supported by a grant from the Swiss National Supercomputing Centre (CSCS) under project ID 520. We would like to thank Martin Werner for providing the ECHAMwiso data, and the IAEA/WMO for providing the GNIP data. Moreover we are grateful to Hui Tang and Emanuel Christner for the help with setting up COSMOiso in climate mode.





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



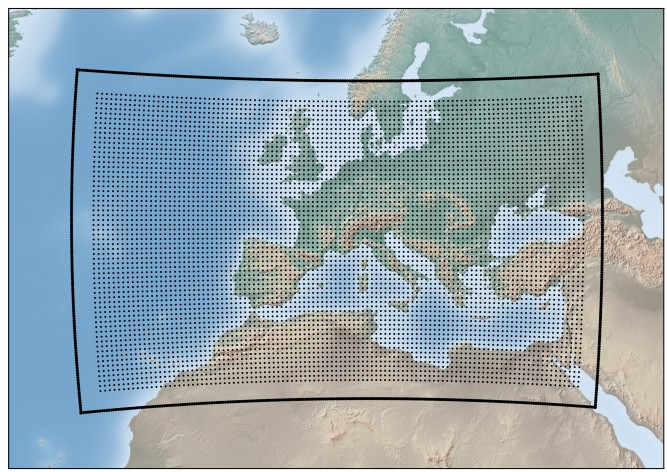

**Figure 1.** Model domain of the COSMOiso simulation (black line) and the starting points of the backward trajectories (dots).

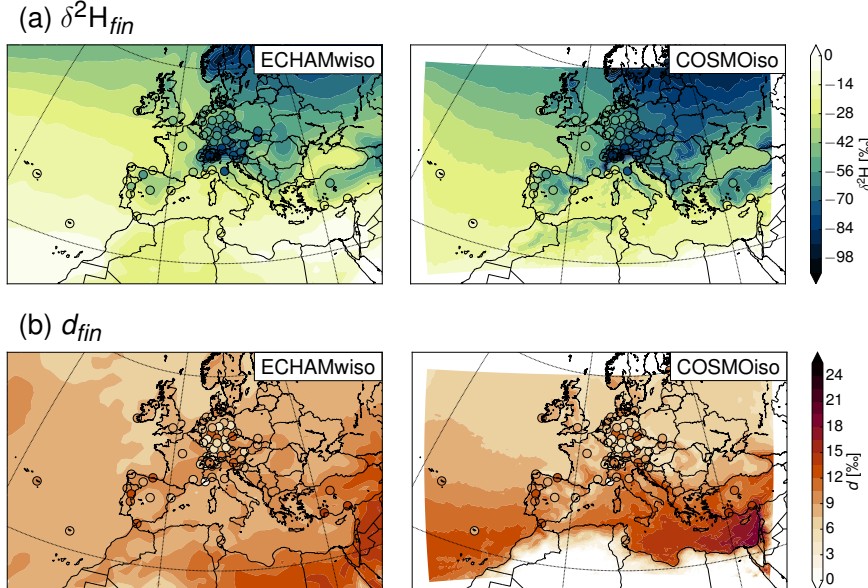

**Figure 2.** 30-year mean of (a) $\delta^2$H and (b) $d$ in precipitation from ECHAMwiso (left) and COSMOiso (right). The GNIP measurements are shown as dots.

Yoshimura, K., Frankenberg, C., Lee, J., Kanamitsu, M., Worden, J., and Röckmann, T.: Comparison of an isotopic atmospheric general circulation model with new quasi-global satellite measurements of water vapor isotopologues, J. Geophys. Res., 116, doi:10.1029/2011JD016035, 2011.





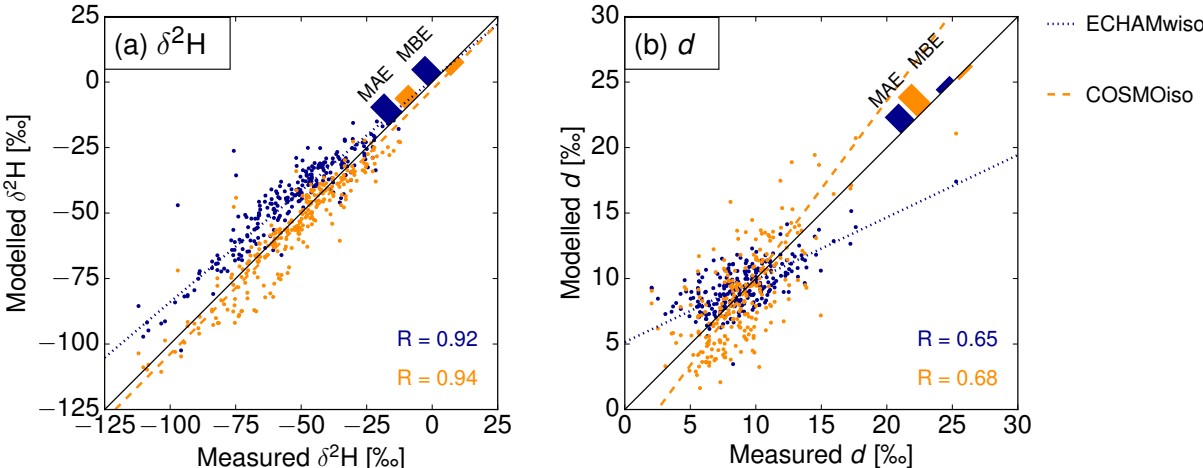

**Figure 3.** Seasonal mean values of $\delta^2$H and $d$ in precipitation from ECHAMwiso (dark blue) and COSMOiso (orange) with respect to the GNIP measurements. Each dot represents the value at one GNIP station averaged over one season and 30 years. The lines are orthogonal regression lines (Adcock, 1878). MAE, MBE and $R$ are the mean absolute error, the mean bias error and the linear correlation coefficient, respectively.

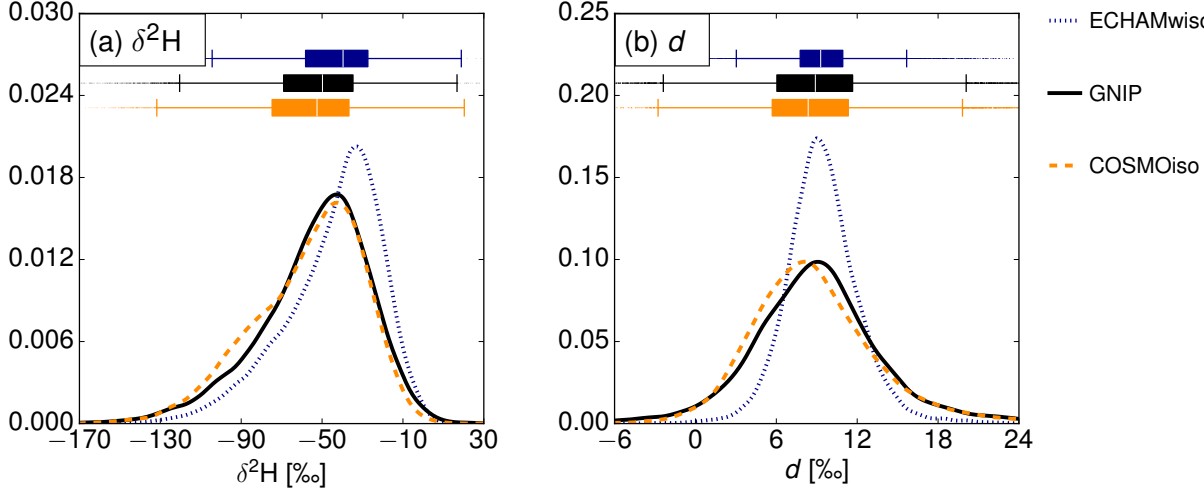

**Figure 4.** Box plots and probability density functions of the monthly mean $\delta^2$H and $d$ in precipitation from ECHAMwiso (dark blue), COSMOiso (orange) and the GNIP measurements (black).

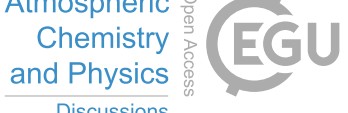



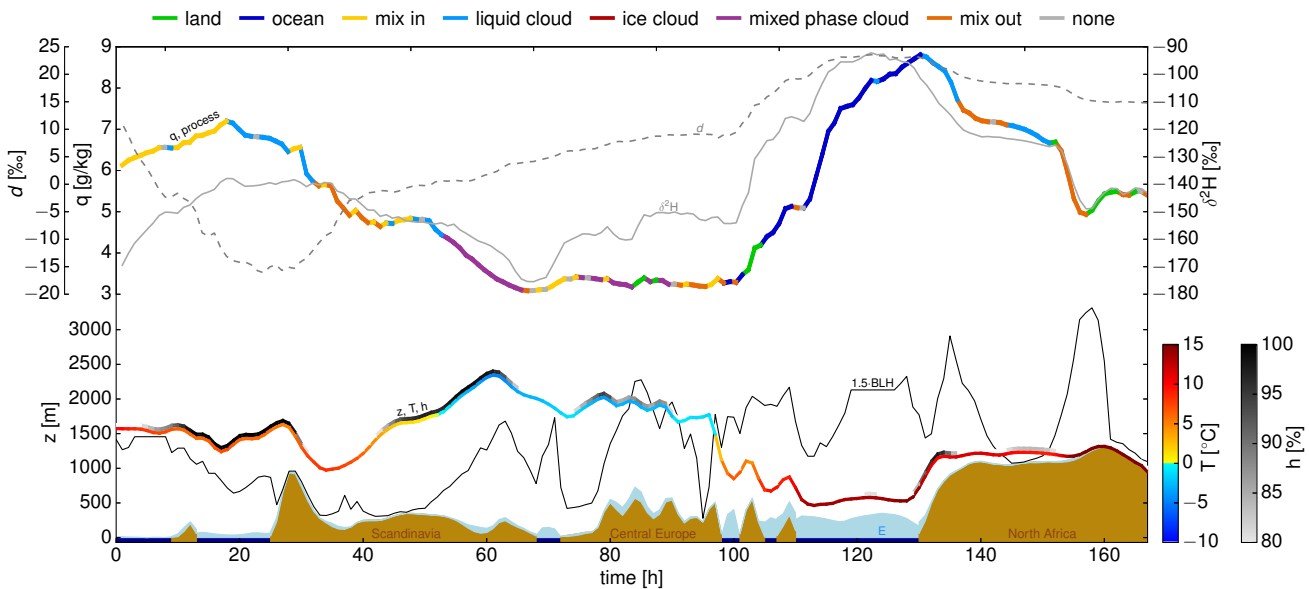

**Figure 5.** Time series of $q$ (coloured with processes, see legend at the top), $\delta^2$H (solid grey line), $d$ (dashed grey line), $z$ (coloured with $T$ and $h$), $1.5 \cdot$ BLH (solid black line), and $E$ (light blue shading, scaled by $10^6$) along an example trajectory arriving in North Africa (33°N, 1°E) on 3 October 1989 at 00 UTC. The brown and blue shadings indicate land and ocean surfaces, respectively.

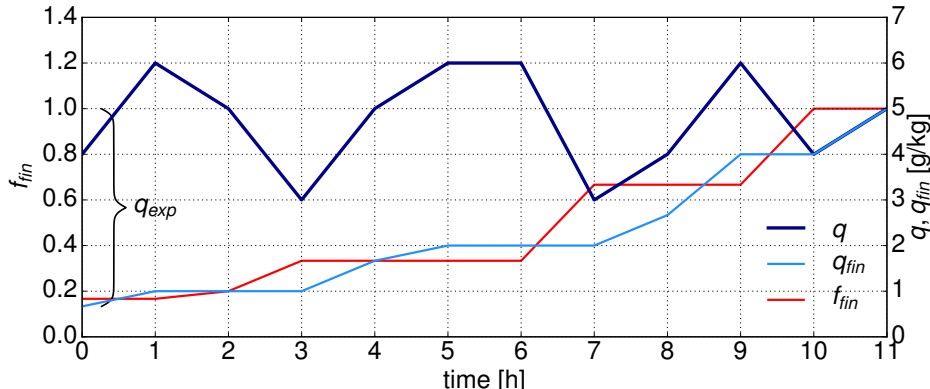

**Figure 6.** Time series of $q$ (dark blue line), $q_{fin}$ (light blue line), and $f_{fin}$ (red line) along a hypothetical trajectory with $N = 11$. The total fraction of explained moisture ($q_{exp}$) corresponds to $1 - q_{fin}^0 / q_{fin}^N$.





**Figure 7.** 30-year mean of $\delta^2$H (left) and deuterium excess (right) in water vapour averaged over the first, third, and fifth lowest model level (top) and the weighted contributions of the initial values (middle) and changes along the trajectories (bottom). The numbers show the mean contribution of each term to the final value ($\delta^2H_{fin}$ and $d_{fin}$). The contours show $1 - q_{exp}$ (middle) and $q_{exp}$ (bottom) in %.





**Figure 8.** Contributions of the processes to $\Delta\delta^2$H in Figure 7 e. The numbers show the mean contribution of each process and the contours show $q_{exp}^k$ (see Equations 10 and 11) in %. Note the different colour scales for $q_{exp}^k$ between the left and right hand side.





**Figure 9.** Contributions of the processes to $\Delta d$ in Figure 7 f. The numbers show the mean contribution of each process and the contours show $q_{exp}^k$ (see Equations 10 and 11) in %. Note the different colour scales for $q_{exp}^k$ between the left and right hand side.





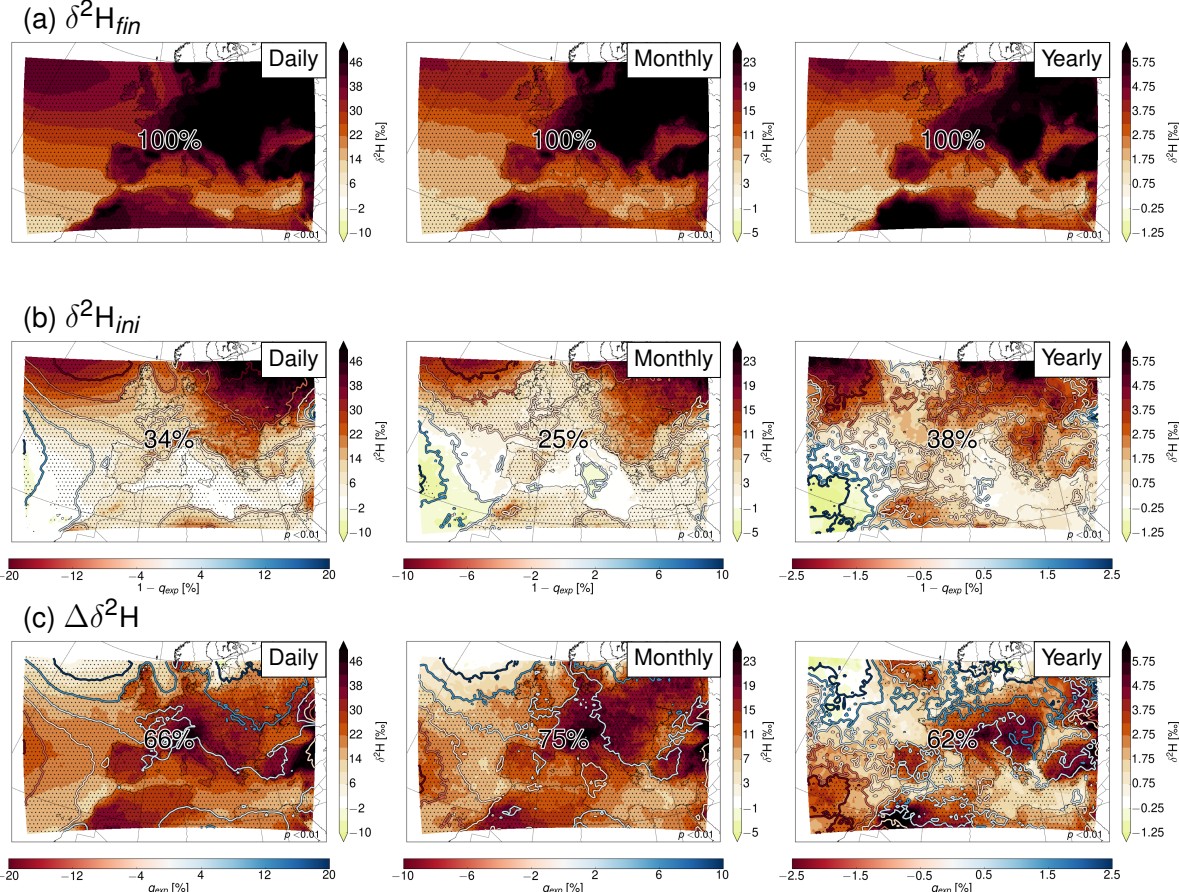

**Figure 10.** Difference in $\delta^2$H between the high and low anomaly days (left), months (middle), and years (right) at each grid point. The top row shows the mean $\delta^2$H in water vapour averaged over the first, third and fifth lowest model level, the middle and bottom rows show the weighted contributions of the initial values and the changes along the trajectories, respectively. The numbers show the mean contribution of each term to $\delta^2$H$_{fin}$, and the contours show the difference in $1 - q_{exp}$ (middle) and $q_{exp}$ (bottom) in %. Stippling indicates areas where $p < 0.01$. Note the different colour scales.







**Figure 11.** Contributions of the processes to the daily $\Delta\delta^2$H in Figure 10 c. The numbers show the mean contribution of each process, and the contours show the difference in $q_{exp}^k$ (see Equations 10 and 11) between the high and low anomaly days in %. Stippling indicates areas where $p < 0.01$. Note the different colour scales for $q_{exp}^k$ between the left and right hand side.





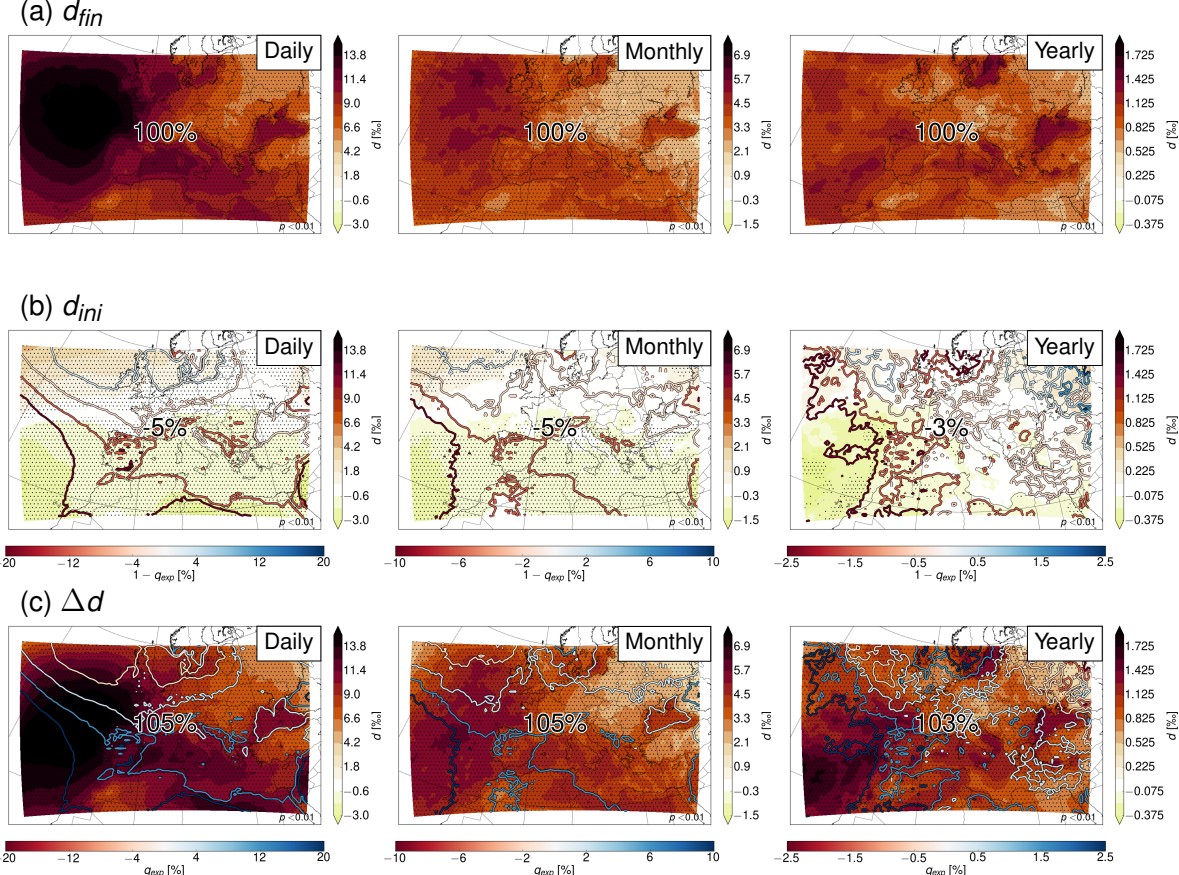

**Figure 12.** Difference in deuterium excess between the high and low anomaly days (left), months (middle), and years (right) at each grid point. The top row shows the mean deuterium excess in water vapour averaged over the first, third and fifth lowest model level, the middle and bottom rows show the weighted contributions of the initial values and the changes along the trajectories, respectively. The numbers show the mean contribution of each term to $d_{fin}$, and the contours show the difference in $1 - q_{exp}$ (middle) and $q_{exp}$ (bottom) in %. Stippling indicates areas where $p < 0.01$. Note the different colour scales.





**Figure 13.** Contributions of the processes to the daily $\Delta d$ in Figure 12 c. The contours show the difference in $q_{exp}^{k}$ (see Equations 10 and 11) between the high and low anomaly days in %. Stippling indicates areas where $p < 0.01$. Note the different colour scales for $q_{exp}^{k}$ between the left and right hand side.