# Peer review of "Lagrangian process attribution of isotopic variations in near-surface water vapour in a 30-year regional climate simulation over Europe"

_Atmospheric Chemistry and Physics, 2017_

## Referee Comment (RC1) · Anonymous Referee #2 · 17 Oct 2017

[english]article     [T1]fontenc     [latin9]inputenc     geometry     verbose,tmargin=2cm,bmargin=2cm,lmargin=2cm,rmargin=2cm babel

[Figure]

**Comments on Dütsch et al**

October 17, 2017

The paper presents a new method to diagnose the contribution from different processes to the isotopic composition of water vapor from model simulations. This method is applied here to a regional model simulation with COSMOiso and to understand the mean and variability of $\delta D$ and d-excess. However, the scope of this paper is actually much broader. The method could be readily applied to any model simulations and in any region of the world. This kind of method is extremely welcome to fill the long-standing gap between the complex numerical simulations (GCM or RCM) and the simple lagrangian models. It can pave the way to improved understanding of isotopic signals in water vapor.

The paper is very well written, the figures are of good quality, the method and associated equations are well explained so that anyone could easily reproduce it with their own model simulations. The method bears limitations, especially the "binary distinction between processes based on thresholds", but these are well identified by the authors and extensively discussed in the discussion section.

Therefore, I recommend acceptation of this paper. I only have very few very minor comments:

- p 4 l 11: add "-" in "terrain-following"?

- p 10 l 13: for daily anomalies, did you subtract the monthly climatological mean? Or does it mix up both day-to-day and seasonal variations?

- p 11 l 15: "deuterium excess from evaporation inside the domain is typically larger than outside the domain": it looks like this rationale would apply to the interpretation of the mean deuterium excess, but not necessarily for it variability. Doesn't it mean that variations in deuterium excess from evaporation inside the domain typically are of opposite sign compared to those outside the domain?

- p 17 fig 3: "seasonal": do you mean monthly?

---

## Referee Comment (RC2) · Anonymous Referee #1 · 17 Oct 2017

General Comments

This paper presents a novel method to diagnose meteorological processes based on isotopes in atmospheric water vapor output from two isotope-enabled climate model. The applications of the methods described here could be far-reaching and apply to other regions of the world, providing a much-needed approach to filling a knowledge gap This paper is well written, and the figures are of high quality to convey the authors' meaning. I recommend accepting this paper for publication after minor issues are clarified.

Specific Comments

p. 4, Line 15 – Could you discuss your rationale for assuming that no fractionation takes place during ET? This discussion could be just a few words with references. For example, do you assume that there is no fractionation because this is a transpiration-dominated system throughout most of the year? On p. 6 line 2, you state that "moisture is assumed to have evaporated from the surface" if q increases during a time step. Why wouldn't fractionation take place during evaporation?

Technical Corrections

p. 5, Line 2 – This sentence is a bit awkward: "At all stations only the months are considered, for which a GNIP measurement and values from both models are available." Consider rewording. (Suggestion: "At all stations, months are included if a GNIP measurement and values from both models are available.")

---

## Author Comment (AC1) · 23 Nov 2017

Manuscript acp-2017-744:

Lagrangian process attribution of isotopic variations in near-surface water vapour in a 30-year regional climate simulation over Europe

**Response to the Reviewers' Comments**

We would like to thank both reviewers for their constructive comments, which helped to clarify the manuscript. In addition to these comments, we included a suggestion from the technical review round and slightly adapted the discussion and conclusions to better illustrate how our method provides new insight compared to previous studies. In the following, we respond point by point to the comments from the interactive discussion. Descriptions of changes we made are highlighted in blue and the corresponding text in the article is shown in green.

**Comments of Reviewer 1:**

1.1) *p. 4, Line 15 - Could you discuss your rationale for assuming that no fractionation takes place during ET? This discussion could be just a few words with references. For example, do you assume that there is no fractionation because this is a transpiration-dominated system throughout most of the year? On p. 6 line 2, you state that "moisture is assumed to have evaporated from the surface" if q increases during a time step. Why wouldn't fractionation take place during evaporation?*
REPLY: Thanks, this is a good point, which indeed requires some explanation. Since the soil isotopes are prescribed by external data, we used the setup without fractionation during evapotranspiration from land to conserve mass (fractionation without feedback on the soil would mean that heavy isotopes are removed from the system). However, we have tested several setups (*Dütsch*, 2016), including one with a coupled isotope soil model (TERRAiso, *Aemisegger*, 2015), which allows for feedbacks between isotopes in the soil and in the atmosphere and distinguishes between bare soil evaporation (with fractionation) and plant transpiration (without fractionation). The difference between the two simulations was small, and the prescribed-soil simulation performed slightly better than the coupled-soil simulation when compared to GNIP measurements. Therefore we

decided to use the setup with prescribed soil isotopes and without fractionation during evapotranspiration. We have extended this paragraph and now mention more specifically which setups have been tested:

This setup has been found to be the best out of six setups with different initial and boundary conditions and different parameterisations of fractionation during ocean evaporation and land evapotranspiration, when compared with monthly isotope measurements in precipitation from the Global Network of Isotopes in Precipitation (GNIP; *IAEA/WMO*, 2016) (*Dütsch*, 2016).

Additionally we added "or transpired" to the sentence on page 6 line 2:

If specific humidity increases during a certain time step ($\Delta q > \Delta q_{min}$), moisture is assumed to have evaporated or transpired from the surface or to be mixed into the air parcel (i.e., the parcel mixes with moister air).

1.2) *p. 5, Line 2 - This sentence is a bit awkward: "At all stations only the months are considered, for which a GNIP measurement and values from both models are available." Consider rewording. (Suggestion: "At all stations, months are included if a GNIP measurement and values from both models are available.")*

REPLY: Rephrased according to your suggestion.

**Comments of Reviewer 2:**

2.1) *p 4 l 11: add "-" in "terrain-following"?*
REPLY: Corrected.

2.2) *p 10 l 13: for daily anomalies, did you subtract the monthly climatological mean? Or does it mix up both day-to-day and seasonal variations?*
REPLY: We subtracted the 31-day running mean of the 30-year daily climatology to avoid seasonal variations. This is mentioned in the text:
The influence of the processes on isotopic variability is addressed by considering days, months and years when $\delta^2$H or deuterium excess are unusually high or low, i.e., when the anomalies with respect to the climatological mean are in the highest or lowest 25 % (33 % for years). ... For the daily anomalies, the climatological mean is calculated as the 31-day running mean of the 30-year daily climatology, ...

2.3) *p 11 l 15: "deuterium excess from evaporation inside the domain is typically larger than outside the domain": it looks like this rationale would apply to the interpretation of the mean deuterium excess, but not necessarily for it variability. Doesn't it mean that variations in deuterium excess from evaporation inside the domain typically are of opposite sign compared to those outside the domain?*
REPLY: This is a good point, thanks. Our reasoning was the following: the negative contribution of the initial deuterium excess values mainly results from the fact that less moisture is explained by the initial values on days with high deuterium excess (lower $1 - q_{exp}$). Since typically the deuterium excess in the moisture evaporated inside the model domain is higher than outside, more moisture taken up inside the model domain could alone lead to a higher deuterium excess. However, we agree that this is a bit confusing, and therefore deleted these two sentences.

2.4) *p 17 fig 3: "seasonal": do you mean monthly?*
REPLY: No, in Figure 3 the dots represent 30-year averaged seasonal mean values. This means that there are (maximum) 4 dots per GNIP station.

**References**

Aemisegger, F. (2015), TERRAiso test case studies using a TERRA stand-alone setup for Rietholzbach, unpublished manuscript.

[revised manuscript text omitted]